# Multi-Omics Analysis of Transcriptomic and Metabolomics Profiles Reveal the Molecular Regulatory Network of Marbling in Early Castrated Holstein Steers

**DOI:** 10.3390/ani12233398

**Published:** 2022-12-02

**Authors:** Fang Sun, Minyu Piao, Xinyue Zhang, Siqi Zhang, Ziheng Wei, Li Liu, Ye Bu, Shanshan Xu, Xiaochuan Zhao, Xiangren Meng, Mengmeng Yue

**Affiliations:** 1Institute of Animal Husbandry, Heilongjiang Academy of Agricultural Sciences, Harbin 150086, China; 2Institute of Feed Research, Chinese Academy of Agricultural Sciences, Beijing 100081, China; 3College of Animal Science and Technology, Northeast Agricultural University, Harbin 150030, China; 4College of Animal Science and Technology, China Agricultural University, Beijing 100193, China; 5College of Animal Science and Veterinary Medicine, Heilongjiang Bayi Agricultural University, Daqing 163319, China

**Keywords:** Holstein steers, early castration, transcriptomic, metabolomics, intramuscular fat

## Abstract

**Simple Summary:**

In this paper, we investigated the early-castrated Holstein cattle liver transcriptome and metabolome and conducted a comprehensive analysis of two omics associated with the IMF deposition using transcriptomics and untargeted metabolomics under different treatments: non−castrated and slaughtered at 16 months of age (GL16), castrated at birth and slaughtered at 16 months of age (YL16), and slaughtered at 26 months of age (YL26).The results demonstrated that implementation of early castration modified the hepatic metabolites and the related biological pathways by regulating the relevant gene expression, which could represent a better rearing method for the production of high-marbled and healthier beef products.

**Abstract:**

The intramuscular fat (IMF), or so-called marbling, is known as potential determinant of the high quality beef in China, Korea, and Japan. Of the methods that affect IMF content in cattle, castration is markedly regarded as an effective and economical way to improve the deposition of IMF but with little attention to its multi-omics in early-castrated cattle. The aim of this study was to investigate the liver transcriptome and metabolome of early-castrated Holstein cattle and conduct a comprehensive analysis of two omics associated with the IMF deposition using transcriptomics and untargeted metabolomics under different treatments: non−castrated and slaughtered at 16 months of age (GL16), castrated at birth and slaughtered at 16 months of age (YL16), and castrated at birth and slaughtered at 26 months of age (YL26). The untargeted metabolome was analyzed using ultrahigh-performance liquid chromatography coupled with quadrupole time-of-flight mass spectrometry. The transcriptome of the hepatic genes was analyzed to identify marbling-related genes. Using untargeted metabolomics, the main altered metabolic pathways in the liver of cattle, including those for lipid and amino acid metabolism, were detected in the YL16 group relative to the GL16 and YL26 groups. Significant increases in the presence of betaine, alanine, and glycerol 3-phosphate were observed in the YL16 group (*p* < 0.05), which might have contributed to the improved beef-marbling production. Compared to the GL16 and YL26 groups, significant increases in the presence of glutathione, acetylcarnitine, and riboflavin but decreases in diethanolamine and 2-hydroxyglutarate were identified in YL16 group (*p* < 0.05), which might have been beneficial to the beef’s enhanced functional quality. The gene expressions of *GLI1* and *NUF2* were downregulated and that of *CYP3A4* was upregulated in the YL16 group; these results were strongly correlated with the alanine, betaine, and leucine, respectively, in the liver of the cattle. In conclusion, implementation of early castration modified the hepatic metabolites and the related biological pathways by regulating the relevant gene expressions, which could represent a better rearing method for production of high marbled and healthier beef products.

## 1. Introduction

With the improvements in people’s living standards, China’s beef consumption has also increased rapidly in recent years. From 2017 to 2019, China’s beef imports amount ranked first in the world [1]. Due to the fact that the shortage of cattle resources is increasingly prominent in the beef cattle industry, the fattening of dairy calves is gradually being given attention by cattle-breeding farms. Since dairy male calf resources are fully utilized in the countries with a developed dairy industry, dairy male cattle are one of the main sources of beef [2]. Some EU countries, especially the Netherlands, mainly use dairy male calves to produce veal; Japan uses Holstein steers to produce marbled beef; and the United States implements castration of dairy male calves to produce high-quality beef. Israel is at the forefront of the world in terms of the fattening technique used for dairy male calves, and the fattening of dairy male calves has become one of the important indices of dairy cattle breeding in Israel [3,4].

Straight-line fattening of calves with free stall rearing will be the main fattening method for the beef cattle industry in the future [5]. Holstein male calves usually begin to ride each other after seven months of age, which not only affects weight gaining, but also leads to the risk of injury, which is inconducive to cattle management. Cattle are usually castrated for various management purposes, among which physiological aid to taming through castration is one [6]; improving beef quality through increased meat marbling is also a reason for castration because it may affect lipid metabolism in the longissimus dorsi of steers [7]. Schaefer [8] reported that approximately 2.35 million Holstein steers are estimated to be marketed annually. With the increased public concern regarding animal welfare, researchers have become increasingly interested in the evaluation of pain in livestock caused by castration and its alleviation method. Ting et al. [9] found that the earlier the castration period of calves, the lower the physiological stress indices such as plasma cortisol, which indicates the calves might suffer less pain caused by castration. However, little information is available on the effects of early castration on animal performance and carcass characteristics in Holstein male calves during the early fattening period. Furthermore, few studies have systematically been reported on the molecular mechanisms of intramuscular fat (IMF) deposition after early castration, which may be attributed to the unknown genes and metabolites of early-castrated Holstein calves.

Marbling responses to castration are related to large-scale changes in metabolism and gene expression [7]. Gene expression profiling of IMF can be conducted by evaluating the transcriptome data sets via bulk RNA-Seq [10]. Transcriptomics provides an effective tool to explore genes that might contribute to IMF deposition. Marbling has diverse highly regulated metabolic pathways that can play significant roles in their production under castration. Metabolomics has also been used in IMF metabolism and biomarker identification [11]. Metabolites that are the final products of diverse biological metabolism serve as precise biomarkers that reflect upstream biological processes including environmental changes and genetic mutations [12]. However, metabolomic biomarkers are known to have low reproducibility due to sample sources, population heterogeneity, parameter settings in the metabolomics data, different experiment protocols, etc. [13]. Currently, as system biology and bioinformatics tools develop and as cost-effective methods for analyzing the linkage between metabolites and gene expression emerge, the combination of metabolomics and transcriptomics data provides a more powerful approach to gaining a comprehensive insight into the mechanisms of IMF-deposition responses to castration at the cellular and molecular levels than either approach alone. This new approach may observe fat deposition as an aspect of system biology and enhance the credibility of biomarkers [14]. Therefore, comprehensive analysis of the metabolomics and transcriptomics can provide a better understanding of marbling responses to castration.

In the current study, Holstein male calves were castrated at birth. We focused on the combination of the liver transcriptome and metabolome to observe the physiological changes and carcass traits of the Holstein cattle. In ruminants, because the liver synthesizes glucose by using propionic acid to regulate the entire energy balance and also regulates fat metabolism via fat accumulation and oxidation [15], we decided to use the liver tissue as the target sample. We first evaluated the effects of early castration on animal performance and marbling of the longissimus muscle in follow-up Holstein growing cattle. Second, to clarify the IMF-deposition mechanisms in the early castrated cattle, an integrated analysis of transcriptomic and metabolomic data was used to evaluate the differences in metabolite biosynthesis and gene expression between bulls and steers. In addition, the correlations between the metabolite biosynthesis and the gene expression were used to gain a better understanding of IMF deposition after castration. We hypothesized that implementation of early castration could affect the marbling level while also alleviating the temperament of the Holstein cattle. This study provided molecular information on the responses of IMF to castration and showed the most related biological pathways that play critical roles in the marbling in early-castrated Holstein cattle.

## 2. Materials and Methods

### 2.1. Animals, Slaughter Procedures, Tissue Sample Collections, and Measurements

Thirty-four Chinese Holstein male calves born from 2 January to 16 February 2018 were reared at Yu Da Dairy Farm, Heilongjiang province in China. Based on their body weight (BW) and age, the calves were assigned to one of two groups in a completely randomized design: the bull group (*n* = 17; served as the control group) and the steer group (*n* = 17; castrated by ligating via rubber band at birth). The experiment was started with an initial BW of about 42 kg, and the periods were 15 months in duration (from 16 February 2018 to 12 May in 2019), in which the suckling period lasted for 60 days and the fattening period lasted for 11 months (from 16 June 2018 to 12 May in 2019) (Figure 1). The animals were reared in the same pen and had free access to diet and water before 1 June 2018. Since 1 June 2018, the animals were separated into a control group pen and a steer group pen, but the animals from the two groups were fed the same diet. The dietary nutrient requirements at different stages of the fattening period were formulated according to the Japanese feed standard for beef cattle (2008, Table 1).

The BW was measured at 09:00 a.m. before feeding at d 1, 120, 260, 400, and 450. Six cattle with a BW similar to the average BW that were 16 months of age were collected from each group, transported for 3 h to a local municipal slaughterhouse on the final experiment day, and then slaughtered following a previously described method [16]. In addition, the remaining cattle were continuously reared until 26 months of age and then slaughtered. Immediately after slaughter, liver samples were taken, frozen in liquid nitrogen, and stored at −80 °C until further analysis, all of which followed our previous sampling procedure [17]. At 24 h post-mortem, the carcasses were evaluated using the beef-quality grading system of the Chinese Agricultural Industry Standard (NY/T 676-2010 2010, China) [18]. At this stage, the marbling score, carcass weight, fat thickness, longissimus muscle (LM) area, fat color, and meat color were examined by an official meat grader. The longissimus thoracis (LT; cold carcass weight: about 500 g) was obtained from the 12th vertebra, vacuum-packaged, and transported on ice to a laboratory. After transportation to the laboratory, the packages containing the LT samples were opened and the external fat was trimmed away. The LT samples were minced using mini chopper (CH180, Kenwood, Shanghai, China) for 30 s. The minced LT samples from various locations were pooled; some samples were used immediately for the evaluation of the pH and chemical composition. Samples for shear force, drip loss, and meat color (CIE value) were collected but not minced, and the shear force, drip loss, and meat color were immediately determined.

The crude fat, crude protein, and moisture were determined according to AOAC methods [19]. The surface-color values (International Commission on Illumination (CIE); *L**, *a**, and *b** values represented lightness, redness, and yellowness, respectively) were measured using a colorimeter (CM-5, Minolta Co., Ltd., Osaka, Japan) as described by Piao et al. [20]. The pH of the beef samples was measured using a pH meter (SevenGo, Mettler-Toledo Inti., Inc., Uster, Switzerland) as described by Piao et al. [21]. The shear force value (N) was measured using a Warner-Bratzler shear machine attached to a texture analyzer (CT3 10K, Brookfield Engineering Laboratories, Middleboro, MA, USA) as described by Piao et al. [21]. The cooking loss was measured using a water bath as described by Piao et al. [21].

### 2.2. Metabolite Extraction

For the metabolomics analysis, we used six samples from each group that were designated as follows: GL16, non−castrated and slaughtered at 16 months of age; YL16, castrated at birth and slaughtered at 16 months of age; and YL26, castrated at birth and slaughtered at 26 months of age. A total of 100 mg of liver sample was added to 250 μL of water, homogenized, and vortexed for 60 s. Then, 1 mL of methanol acetonitrile solution (1:1, *v*/*v*) was added, and the mixture was vortexed for 1 min and ultrasonicated for 30 min twice. The protein was precipitated at −20 °C for 1 h and centrifuged at 14,000× *g* for 20 min. The supernatant was used for ultrahigh-performance liquid chromatography coupled with quadrupole time-of-flight mass spectrometry (UHPLC-QTOF-MS) analysis.

### 2.3. UHPLC-QTOF-MS Analysis and Data Processing

An Agilent 1290 Infinity LC HPLC HILIC column was used for the separation of metabolites as described in Wang et al. [22]. The column temperature was 25 °C and the flow rate was 0.3 mL/min. The mobile phase was composed of A: water + 25 mM ammonium acetate + 25 mM ammonia water; and B: acetonitrile. The samples were placed in an automatic sampler at 4 °C during the entire analysis process. To avoid the influence caused by the fluctuation of the instrument’s detection signal, a random sequence was used for the continuous analysis of samples. Quality control (QC) samples were prepared based on the mixed liver samples in equal quantities. To assess the reproducibility of this experiment, the samples of each group were mixed into the QC samples in equal quantities. The analysis results of the QC samples were compared for overlapping spectra and showed that the response intensity and retention time of each chromatographic peak overlapped substantially, which indicated that the experiment was relatively repetitive. A principal component analysis (PCA) was performed on all experimental samples and QC samples after Pareto scaling. The QC samples were closely clustered and located in the middle of each group, which indicated that the metabolomics analysis showed a high reproducibility. Electrospray ionization (ESI) in positive and negative ion modes were used for the metabolite detection. The ESI conditions were as follows: ion source gas 1 (gas 1): 60; ion source gas 2 (gas 2): 60; curtain gas (CUR): 30; source temperature: 600 °C; ion spray voltage floating (ISVF): ±5500 V (positive and negative modes); TOF MS scan m/z range: 60–1000 Da; product ion scan m/z range: 25–1000 Da; TOF MS scan accumulation time: 0.20 s/spectra; and product ion scan accumulation time: 0.05 s/spectra. The secondary mass spectra were obtained via information-dependent acquisition (IDA) in high-sensitivity mode with a declustering potential (DP) of ±60 V (positive and negative modes) and a collision energy of 35 ± 15 eV. The IDA conditions were as follows: isotopes excluded within 4 Da and 6 candidate ions to monitor per cycle.

An AB Triple TOF 6600 mass spectrometer was used to identify the metabolites based on the collected primary and secondary spectra of the QC samples. The original data were converted into mzXML format using ProteoWizard (Proteome Software, Inc., Portland, OR, USA), and then the XCMS program (Version 3.7.1) was used for the peak alignment, retention time correction, and peak area extraction. The metabolite structure identification was based on accurate mass-matching (<25 ppm) and secondary-spectrum-matching methods and a search of the laboratory’s self-built commercial database (the quantity was 30,000+; 6000+ metabolites were self-built with standard products including 4000+ for human and animal and 2000+ for plant). The others were from four public databases: Mass Bank, Metlin, HMDB, and MoNA corresponding secondary-standard spectrum library. After the data were preprocessed via Pareto scaling, a multi-dimensional statistical analysis was performed that included an unsupervised principal component analysis (PCA), a supervised partial-least-squares discriminant analysis (PLS-DA), and an orthogonal partial-least-squares discriminant analysis (OPLS-DA). The first principal component of the variable importance in the projection (VIP) was obtained from the OPLS-DA to refine this analysis. The metabolites with VIP values exceeding 1 and the variables assessed via a Student’s *t*-test with a *p*-value < 0.05 were set as the differential metabolites (DFMs). The fold-change value of each metabolite was calculated by comparing the mean value between GL16 and YL16, GL16 and YL26, and YL16 and YL26. The DFMs were further identified and validated using the Kyoto Encyclopedia of Genes and Genomes (KEGG; http://www.genome.jp/kegg/pathway.html, accessed on 7 July 2020). The KEGG library was applied in the enrichment analysis of the KEGG metabolic pathway based on the DFMs. Fisher’s exact test was used to analyze and calculate the significance level of the enrichment pathway; the threshold for the Fisher’s test was *p* ≤ 0.05, and tendencies were indicated by 0.05 < *p* ≤ 0.10.

### 2.4. RNA Extraction and Library Preparation

#### 2.4.1. RNA Extraction

The total RNA was extracted from the liver tissue using TRIzol^®^ Reagent (Magen R4801-02) according the manufacturer’s instructions (Magen, Guangzhou, China). RNA samples were detected based on the A260/A280 absorbance ratio with a Nanodrop ND-2000 system (Thermo Fisher Scientific Inc., Waltham, MA, USA), and the RIN of RNA was determined using an Agilent Bioanalyzer 4150 system (Agilent Technologies Inc., Santa Clara, CA, USA). Only qualified samples were used for library construction.

#### 2.4.2. Library Preparation and Sequencing

Paired-end libraries were prepared using a ABclonal mRNA-seq Lib Prep Kit (ABclonal Technology Co., Ltd., Wuhan, China) following the manufacturer’s instructions. The mRNA was purified from 1 μg of total RNA using oligo (dT) magnetic beads followed by fragmentation carried out using divalent cations at elevated temperatures in ABclonal First Strand Synthesis Reaction Buffer. Subsequently, first-strand cDNAs were synthesized with random hexamer primers and reverse transcriptase (RNase H) using mRNA fragments as templates followed by second-strand cDNA synthesis using DNA polymerase I, RNAseH, buffer, and dNTPs. The synthesized double-stranded cDNA fragments were then adapter-ligated for preparation of the paired-end library. The adaptor-ligated cDNA fragments were used for the PCR amplification. The PCR products were purified (AMPure XP system), and the library quality was assessed on an Agilent Bioanalyzer 4150 system. Finally, the sequencing was performed with an Illumina Novaseq 6000/MGISEQ-T7 (Illumina/BGI, San Diego, CA, USA) instrument.

### 2.5. Data Analysis

The data generated from Illumina/BGI platform were used for a bioinformatics analysis. All of the analyses were performed using an in-house pipeline from Shanghai Applied Protein Technology Inc. The major software and parameters were as follows. The raw data (or raw reads) in fastq format were firstly processed through in-house Perl scripts. In this step, we removed the adapter sequence and filtered out low-quality (when the number of lines with a string quality value less than or equal to 25 accounted for more than 60% of the entire reading) and N (when the base information could not be determined) reads with a ratio greater than 5% to obtain clean reads that could be used for the subsequent analysis. Then, the clean reads were separately aligned to the reference genome in orientation mode using HISAT2 software (Version 2.1.0; http://daehwankimlab.github.io/hisat2, accessed on 3 August 2020) to obtain the mapped reads. The mapped reads were spliced using Stringtie software (http://ccb.jhu.edu/software/stringtie, accessed on 3 August 2020), and then we used the Gffcompare software (http://ccb.jhu.edu/software/stringtie/gffcompare.shtml, accessed on 3 August 2020) to compare them with the reference genome’s GTF/GFF file to find the original unannotated transcription region and discover new transcripts and new genes of the species. FeatureCounts (Version 2.0.0; http://subread.sourceforge.net, accessed on 3 August 2020) was used to count the read numbers mapped to each gene. Then, the FPKM of each gene was calculated based on the length of the gene and the read count mapped to this gene. A differential expression analysis was performed using DESeq2 (http://bioconductor.org/packages/release/bioc/html/DESeq2.html, accessed on 3 August 2020); DEGs with |log2FC|>1 and Padj < 0.05 were considered to be significantly different expressed genes. The GO and KEGG enrichment analysis of the differential genes could explain the functional enrichment of differential genes and clarify the differences between the samples at the gene-function level. We used the topGO package (version of topGO: 2.46.0; GO.db: 3.14.0) for the GO function enrichment and KEGG pathway enrichment analyses. When *p* < 0.05, we considered the GO or KEGG function to be significantly enriched. A transcription factor is a type of protein that can bind to specific DNA sequences that exist widely in organisms. It can be recognized and bound to the cis-acting elements in the upstream regulatory region of genes through specific functional domains, which can activate or hinder the expression of genes. The TF analysis of the DEGs was extracted directly from the AnimalTFDB (http://bioinfo.life.hust.edu.cn/AnimalTFDB, accessed on 18 October 2018)/PlantTFDB database (http://planttfdb.cbi.pku.edu.cn, accessed on 18 October 2018). For the species that did not exist in the database, we first annotated the genes in the Pfam database (http://pfam.xfam.org, accessed on 18 October 2018) using the interscan tool to find the genes and the DBD database (https://dblp.uni-trier.de/rec/journals/nar/KummerfeldT06.html, accessed on 18 October 2018) to screen for the transcription factors. A PPI analysis was used to study whether there was any interaction between the gene products (proteins). The analysis was based on the protein information corresponding to the genes and used the STRING database (https://www.string-db.org, accessed on 18 December 2018), which contains known and predicted protein–protein interactions. For the species existing in the database, we constructed the networks by extracting the target gene list from the database. For species not included in the database, the target gene set sequence was firstly aligned with the reference sample protein sequence contained in the STRING protein interaction database using blastx, and then the protein interaction relationship of the reference species was used to establish an interaction network. Alternative splicing is an important mechanism for regulating the expression of genes and the variability of proteins. rMATS (http://rnaseq-mats.sourceforge.net/index.html, accessed on 3 August 2020) is a variable splicing analysis software suitable for RNA-Seq data. It can not only classify variable splicing events, but also classify deformable splicing events as follows: skipped exon (SE), alternative 5 splice site (A5SS), alternative 3 splice site (A3SS), mutually exclusion exon (MXE), and retained intron (RI).

### 2.6. Statistical Analysis

The growth performance, carcass characteristics, and chemical and physio-chemical composition data were obtained by using the general linear model procedure (Proc GLM) in the SAS 9.3 software (SAS Institute, Cary, NC, USA). The threshold for significance was *p* ≤ 0.05, and tendencies were indicated by 0.05 < *p* ≤ 0.10. The MVDA was performed on differentially expressed abundant genes and metabolites using SIMCA version 14.1. The differentially abundant proteins/modified peptides/genes and metabolites/lipids were log2-scaled (TMT/iTRAQ) or Z-score-scaled (label-free) and concatenated into one matrix. Then, the correlation coefficients among all the molecules in the matrix were calculated using the Pearson algorithm in R version 3.5.1.

## 3. Results

### 3.1. Growth Performance and Carcass Characteristics

The effect of early weaning on growth performance is presented in Figure 2. The initial BW of the beef calves did not differ between the bulls and steers, while the BW of the steers was lower than that of the bulls at d 120 (*p* = 0.024), d 180 (*p* = 0.003), and d 260 (*p* = 0.017). However, the BW did not differ between the bull and steers at d 400 (*p* = 0.393) and d 450 (*p* = 0.246). The effects of early weaning on the carcass characteristics and chemical and physio-chemical compositions are presented in Table 2 and Table 3. The marbling score (*p* = 0.0001) and backfat thickness (*p* = 0.001) were higher in steers than those in bulls, while the LM area of bulls showed the trend to be higher than that of steers (*p* = 0.053). The crude fat content of LT was higher in steers than that in bulls (*p* = 0.008), while the moisture (*p* = 0.75) and crude protein (*p* = 0.17) contents did not differ between the two groups. In addition, the physio-chemical compositions also did not differ (*p* > 0.05) between the two groups.

### 3.2. Analysis of RNA Deep-Sequencing Data

In this study, 40,147,578 to 47,247,592 clean reads of 100 bp for each sample were obtained (Appendix A). Approximately 97% of the clean reads could be mapped to cattle chromosomes, and approximately 95% of the reads in each sample were uniquely mapped to the cattle genome. The number of multiple mapped reads was <2.54% (Appendix A). The Pearson correlation analysis showed that the FPKM values between the three groups were highly correlated (R^2^ = 0.92–0.96; Appendix A).

### 3.3. Identification of Differentially Expressed Genes

The total number of genes expressed in the liver ranged from 12,548 to 13,681 with the numbers of expressed genes being similar among the three groups. Correlations between the biological replicate samples showed a high reproducibility of the expressed genes, which indicated that a major fraction of the liver transcriptome was conserved between the groups. To better explore the biological mechanism of early castration on intramuscular fat deposition, the identification of differentially expressed genes between the two different stages was very important.

The visualized profiles of DEGs in this study are shown in Appendix A. Compared to the non−castrated cattle in GL16 group, a total of 59 DEGs were found in the liver tissue of YL16 cattle including 25 downregulated and 34 upregulated DEGs, which accounted for 42.4% and 57.6% of the total DEGs, respectively. After a fattening period of 10 months, 85 DEGs of the YL26 group were detected in the cattle livers compared to the YL16 group, of which 28 were upregulated and 57 were downregulated.

### 3.4. Functional Enrichment Analysis of the DEGs

The DEGs were categorized according to their cellular components (CCs), molecular functions (MFs) and biological processes (BPs) (Figure 3). The DEGs between GL16 and YL16 were shown to be associated with biological processes such as the positive regulation of nervous system development (GO: 0051962), the regulation of developmental growth (GO: 0048638), the positive regulation of cell death (GO: 0010942), and the regulation of neuron differentiation (GO: 0045664). The molecular functions of the annotated proteins were mainly related to oxidoreductase activity acting on paired donors, with incorporation or reduction of molecular oxygen; reduced flavin or flavoprotein as one donor; and incorporation of one atom oxygen (GO: 0016712), monooxygenase activity (GO: 0004497), JAK pathway signal transduction adaptor activity (GO: 0008269), and linoleoyl-CoA desaturase activity (GO: 0016213). The DEGs between YL16 and YL26 were shown to be associated with biological processes such as stem cell division (GO: 0017145), the steroid biosynthetic process (GO: 0006694), angiotensin-mediated drinking behavior (GO: 0003051), and the negative regulation of interleukin-10 biosynthetic process (GO: 0045081). The molecular functions of the annotated proteins were mainly related to type II activin receptor binding (GO: 0070699), hydroxymethylglutaryl-CoA synthase activity (GO: 0004421), and proline dehydrogenase activity (GO: 0004657).

The KEGG pathway enrichment and protein–protein interaction network (PPI) analyses were used to determine the over-represented biological events and to provide a primary overview of the liver transcriptome that was affected by early weaning. The pathway enrichment analysis between GL16 and YL16 showed that 21 differentially expressed genes participated in 44 pathways (Appendix A). Among the 44 pathways that were identified, 6 upregulated genes were enriched in 13 pathways, and 15 downregulated DEGs were mapped to 31 KEGG pathways, including the NF-kappa B signaling pathway, chronic myeloid leukemia, small cell lung cancer, linoleic acid metabolism, thyroid cancer, etc. A total of 21 DEGs between YL16 and YL26 participated in 21 pathways (Appendix A). Among the 21 pathways, 13 upregulated genes were enriched in 9 pathways and 8 downregulated genes were mapped to 12 pathways, including glycerolipid metabolism, synthesis and degradation of ketone bodies, terpenoid backbone biosynthesis, the renin–angiotensin system, and butanoate metabolism.

### 3.5. Beef Hepatic Metabolome

The PCA plots and volcano plots of the positive and negative ion modes for the three groups are shown in Figure 4. A good separation of the liver metabolites among the three groups was achieved in the PCA score plots of the positive ion mode. The parameters for the assessment of the PCA and OPLS-DA model quality in discriminating every two comparisons among the three groups can be seen in Appendix A, and the OPLS-DA score plots are indicated in Appendix A. The OPLS-DA analysis showed a clear distinction between GL16 and YL16 in both the positive (R2Ycum = 0.987, Q2cum = 0.707) and negative modes (R2Ycum = 0.995, Q2cum = 0.731), which was validated by the permutation analysis (positive: Q2 intercept = −0.075; negative: Q2 intercept = −0.038). The OPLS-DA analysis showed a clear distinction between YL16 and YL26 in both the positive (R2Ycum = 0.99, Q2cum = 0.339) and negative modes (R2Ycum = 0.984, Q2cum = 0.211), which was validated by the permutation analysis (positive: Q2 intercept = 0.052; negative: Q2 intercept = 0.075). The OPLS-DA analysis revealed a clear distinction between GL16 and YL26 in both the positive (R2Ycum = 0.995, Q2cum = 0.76) and negative modes (R2Ycum = 0.996, Q2cum = 0.677), which was validated by the permutation analysis (positive: Q2 intercept = 0.024; negative: Q2 intercept = 0.053). Based on the cutoff (VIP > 1 and *p* < 0.05) for DFMs, 131 DFMs (81 in the positive mode and 50 in the negative mode) were identified in the comparison between GL16 and YL16, 118 DFMs (72 in the positive mode and 46 in the negative mode) were identified in the comparison between YL16 and YL26, and 112 DFMs (65 in the positive mode and 47 in the negative mode) were identified in the comparison between GL16 and YL26 (Appendix A).

The KEGG pathway enrichment analysis revealed the top 20 pathways that were influenced by early castration (Figure 5). Among them, most metabolic pathways belonged to lipid metabolism, protein metabolism, and mineral metabolism. As shown in Table 4, ABC (ATP-binding cassette) transporters, protein digestion and absorption, and biosynthesis of amino acids were the mutual different pathways among the comparisons of the three groups. Among these key metabolic pathways, the main DFMs including glutamate, leucine, maltose, cystine, and tyrosine were upregulated; and glutamine, betaine, and 2-oxoadipic acid were downregulated in the GL16 group compared to YL16 and YL26 (*p* < 0.05).

### 3.6. Correlation Analysis

We found that the DFMs were correlated with the DEGs (Figure 6). In the comparison between GL16 and YL16, gamma-L-glutamyl-L-glutamic acid was negatively correlated with *AKR1B10* (aldo-keto reductase family 1 member B 10). The 5-L-glutamyl-L-alanine and L-alanine were negatively correlated with *GLI1* (GLI family zinc finger 1). The 16-hydroxypalmic acid was positively correlated with *FADS2* (fatty acid desaturase 2) and *CYP3A4* (cytochrome P450 subfamily IIIA polypeptide 4). DL-2,4-diaminobutyric acid and L-pyroglutamic acid were positively correlated with *AHNAK* (AHNAK nucleoprotein). Riboflavin was positively correlated with *CLRN2* (clarin 2) and *LDLRAD3* (low-density lipoprotein receptor class A domain-containing 3). Betaine was negatively correlated with *NUF2* (NUF2 component of NDC80 kinetochore complex). Xanthosine was positively correlated with *GADD45G* (growth arrest and DNA damage-inducible gamma) and negatively correlated with *NPR3* (natriuretic peptide receptor 3). L-serine was positively correlated with *SFXN1* (sideroflexin 1) and *MACROD2* (mono-ADP ribosylhydrolase 2). The 1-methylguanosine and L-asparagine were positively correlated with *DUSP6* (dual-specificity phosphatase 6) and negatively correlated with *ZNF286A* (zinc finger protein 286A) and *EPHA6* (EPH receptor A6). Glycine was negatively correlated with *CLRN2* (clarin 2) and *LDLRAD3* (low-density lipoprotein receptor class A domain-containing 3). L-leucine was negatively correlated with *FADS2* (fatty acid desaturase 2) and *CYP3A4* (cytochrome P450 subfamily IIIA polypeptide 4). L-threonine was negatively correlated with *AHNAK* (AHNAK nucleoprotein). D-pipecolinic acid and MG were positively correlated with *CUX2* (cut-like homeobox 2) and negatively correlated with *GADD45B* (growth arrest and DNA damage-inducible beta). NG-dimethyl-L-arginine, 1,3-dimethyluric acid, and DL-3-hydrobutyric acid were negatively correlated with *CXCL3* (chemokine (C-X-C motif) ligand 3) and *NEURL3* (neutralized E3 ubiquitin protein ligase 3). (S)-2-hydroxyglutarate and thioetheramide-PC were negatively correlated with *SPACA3* (sperm acrosome associated 3). Diethanolamine and (-)-riboflavin were positively correlated with *GLI1* (GLI family zinc finger 1). Gama-L-glutamyl-L-phenylalanine and taurocholate were negatively correlated with *CYP2E1* (cytochrome P450 family 2 subfamily E polypeptide 1).

## 4. Discussion

The hypothesis that the early castration could influence the marbling level while alleviating the temperament of the Holstein cattle was supported by the results of this study. The present study demonstrated that implementing the castration of calves caused a testosterone deficiency during the subsequent growth period, which predisposed the calves to deregulate lipid and glucose metabolism, leading to increased adipose content in the liver and peripheral tissues. The present and our previous findings provided a scientific basis for monitoring the effect of early castration on IMF deposition and relationship between metabolites and genes related to lipid and glucose metabolism in cattle for the entire growth period.

Implementation of the castration of cattle is known to improve temperament and meat quality, while it might restrain an animal’s development, muscularity, and meat yield due to the decreased anabolic properties of testosterone [23]. In this study, the BWs of the cattle were lower for steers than those in bulls at d 120, 180, and 260, while they did not differ at d 400 and 450. This was consistent with a previous study [24] in which the final BW and average daily gain of feedlot-finished Nellore cattle were lower for those that were castrated than for those that were non−castrated. Freitas et al. [25] also reported that the castrated feedlot-finished crossbred cattle had a lower total weight gain and carcass yield than non−castrated cattle. In terms of the lower growth rate in castrated cattle relative to non−castrated cattle, the results could be attributed to the absence of the anabolic effect of testosterone because this hormone can increase lean tissue deposition and further growth traits as well as feed efficiency [24,26]. Thus, it was indicated that castration treatment was negatively correlated with the average daily gain and feed efficiency.

The intramuscular fat, namely marbling, is known as the key factor in evaluating beef quality because marbling is positively associated with some of the sensory traits in beef [21]. In the current study, the marbling score and backfat thickness of steers and the crude fat content in their LTs were higher than those of bulls. Li et al. [27] reported that castration increased the IMF content in the psoas major muscle of castrated lambs, which may have been due to the effects of testosterone on lipid accumulation. Supporting the present results, Mach et al. [28] also found that the IMF percentage in the LT of castrated Holstein cattle that were fed high-concentrate diets was higher than that of non−castrated cattle, which may have been due to testicular function suppression. In addition, other previous studies reported that castration increased the IMF content of the longissimus muscle and backfat thickness in steers regardless of the castration time [29,30], which was consistent with the results of the current study. The findings were in agreement with the principle that testosterone deficiency contributes to the increment in lipid droplets in skeletal muscle [31] and improves adipocyte differentiation rather than that of myocytes in pluripotent mesenchymal stem cells through androgen-receptor-dependent signaling [32]. Moreover, a high content of testosterone inhibited preadipocyte preparation in and between intramuscular bundle in stromal vascular cells [33]. Therefore, castration would still be an effective way to produce meat with a high level of marbling.

Next-generation sequencing technology has tremendously empowered researchers in searching for insights into biological mechanisms to investigate the changed phenotype of animals. In the current study, RNA-Seq was used to evaluate the liver transcriptome changes in cattle in response to early castration. In this study, 97.25% of the clean reads could be mapped to the bovine genome, and the number of reads with multiple matches was below 2.54%. The analysis of the Pearson correlation coefficients between the three groups showed a correlation coefficient of 0.96 among the samples, which suggested that the transcriptome analysis results were highly reliable.

The liver is a major organ for nutrient metabolism that detoxifies various metabolites, synthesizes proteins, and produces biochemicals necessary for digestion and growth. The GO enrichment analysis between GL16 and YL16 showed that DEGs involved in biological processes were mainly implicated in the positive regulation of nervous system development and regulation of developmental growth; the related genes (*GLI1*, *IRX3*, *CUX2*, *DUSP6*, et al.) were downregulated in early-castrated animals, which indicated that castration imparted a negative impact on the animals’ growth performance. This finding was similar to that of a previous study in which castration restricted the muscle growth and further body weight in mice [34]. The protein–protein interaction network analysis showed that DEGs enriched in the NF-kappa B signaling pathway were also associated with the regulation of growth and apoptosis. In addition, the previous studies also reported that these development-related genes were closely connected with animal development, which indicated that they functioned as the regulators of cell determination, specification, and proliferation [35,36].

The liver converts dietary nutrients into the fuels and precursors necessary for peripheral tissues and transports them via the blood. A previous study reported that implementation of the castration of Korean beef cattle led to some changes in the metabolism in the liver [37]. Similar findings were also reported in another study in which castration-induced testosterone deficiency affected the body adiposity and mRNA expression of genes associated with lipid metabolism and glucose uptake in hepatic and extrahepatic tissues such as the skeletal muscle, subcutaneous fat, and abdominal fat [38]. The results of the KEGG analysis in the current study indicated that the pathways related to the metabolism of lipids, such as linoleic acid metabolism and steroid hormone biosynthesis, were upregulated after early castration. In these two lipid-metabolism-related pathways, *CYP2E1* and *CYP3A4* were detected, and the expression of two genes were also upregulated in the YL16 group.

Based on untargeted metabolomics, we uncovered enriched pathways for protein digestion and absorption, biosynthesis of amino acids, alanine, aspartate and glutamate metabolism, purine metabolism, glycerophospholipid metabolism, and primary bile acid biosynthesis in the YL16 group. In these pathways, the biosynthesis of glycerol 3-phosphate and alanine were upregulated while those of xanthosine, glycine, serine, and taurocholate were downregulated in the YL16 group compared to GL16 group, which was connected to fat deposition. Furthermore, based on the LC-MS/MS untargeted metabolomics, other metabolites associated with the meat function or sensory traits were detected. In the pathways of ABC transporters and glycerophospholipid metabolisms, betaine and glycerol 3-phosphate were upregulated in the YL16 group. Betaine, namely trimethyl glycine, is a product of choline oxidation and serves as a source of labile methyl groups [39]. Betaine is related to lipoid metabolism via its lipotropic activity, potentially via a sparing effect of choline for labile methyl groups. Then, the spared choline can be used to synthesize lecithin, which is used for fat transport [40]. Glycerol 3-phosphate is produced from glycerol through catalysis of glycerol kinase, which can induce gluconeogenesis, thereby carrying reducing equivalents from the cytosol to mitochondria for oxidative phosphorylation and further acting as the backbone of glyceride lipids [41]. We also found that the glutathione, acetylcarnitine, riboflavin, and alanine were enhanced in the YL16 group. Glutathione is known to act as a major antioxidant to protect organisms from oxidative damage [42]. Acetylcarnitine is formed by the conjugation of free carnitine and acetyl-CoA mediated through enzyme carnitine acyltransferase, and its formation is important in maintaining metabolic flexibility, which leads to improved glucose homeostasis [43,44]. Acetylcarnitine, which has neurotrophic, neuroprotective, and neuromodulatory properties, can play a critical role in preventing various disease processes [45]. Riboflavin, one of the essential water-soluble vitamins for humans, mainly exists in the form of flavin mononucleotide (FMN) and the flavin adenine dinucleotide (FAD) coenzyme, which are involved in redox reactions and also are related to energy and lipid metabolism [46]. Riboflavin is also known to not only reduce blood lipid content and inhibit platelet aggregation and lipid peroxidation, but also promote anti-carcinogenesis and diuresis [47]. Alanine, which is an aliphatic amino acid, is known to be more concentrated in tender beef during post-mortem aging [48], and it can contribute to sweetness along with glutamine and glycine [49]. Thus, the increased levels of these metabolites in the YL16 group hinted at the potential enhanced functional and sensory quality of early-castrated cattle. Regarding the GL16-group-related DFMs, diethanolamine and 2-hydroxyglutarate are potentially toxic and deleterious compounds. Diethanolamine, an alkanolamine, is used in various industrial processes including cosmetic formulations, detergents, surfactants, and pharmaceuticals. Previous studies conducted in rodents reported an relationship between the long-term exposure to diethanolamine and liver/kidney tumors [50] and that dermal administration of diethanolamine to pregnant mice altered the development of the fetal mouse hippocampus via a reduced proliferation of neuronal precursor cells and increased apoptosis [51]. 2-Hydroxyglutarate is known to be an “oncometabolite” that plays a certain role in driving malignant phenotypes and serves as a potential disease marker [52,53]. Thus, the increased level of these metabolites in the GL16 group’s livers might have potentially negative impacts on human health. Based on the targeted IMF determination and untargeted metabolomics studies, we found that the more functional compounds were enhanced in the YL16 group, which in turn might have improved the functional quality of the beef.

A comprehensive analysis of the metabolomics and transcriptomics of the liver tissue was also conducted in this study, in which the correlations between metabolites and genes were analyzed. In the correlation analysis between GL16 and YL16, the L-alanine content was negatively correlated with *GLI1* gene expression. L-alanine is known to be associated with beneficial meat sensory traits such as tenderness and flavor. *GLI1* and *GLI2* act as a transcription factors in the hedgehog signaling pathway, and *GLI1* is a target gene of this pathway [54]. The hedgehog signaling pathway is known to improve the differentiation of mesenchymal stem cells into osteoblasts [55] and chondrocytes [56]; however, it restrained adipocyte differentiation [57]. Previous studies also reported that the hedgehog signaling pathway improved myogenesis in mice [58] and chicks [59], which indicated that the upregulation of this pathway may not be beneficial to meat tenderness. Thus, the enhanced expression of the *GLI1* gene in the GL16 group may have promoted myogenesis, which may have decreased the beef tenderness in the bulls, while the enhanced content of alanine in the YL16 group may have been beneficial to the beef sensory traits of the steers. Betaine content showed a negative correlation with the expression of the *NUF2* gene. Betaine is known to be associated with lipid metabolism as a product of choline oxidation due to the fact that choline is implicated in lecithin synthesis as described above. *NUF2*, also called *CDCA1*, is one of the components of the NDC80 complex that is important in the stabilization of kinetochore–microtubule anchoring and in supporting the centromeric tension involved in the establishment of correct chromosome congression [60]. The *NUF2* gene is also known to be consistently overexpressed in human hepatocellular carcinoma tissues; its sufficient expression ensured the appropriate growth of hepatocellular carcinoma cells [61], which may have a negative impact on animal health. Thus, although there is little information on the relationship between betaine and the *NUF2* gene, the enhanced content of betaine in the YL16 group may have activated fat metabolism in the body, and the downregulated expression of *NUF2* may have indicated a healthier state of the cattle in YL16 group. In addition, the diethanolamine content showed a positive correlation with the expression of the *GLI1* gene. As mentioned above, long-term exposure to diethanolamine is associated with liver/kidney tumors. Thus, the enhanced content of diethanolamine in GL16 may have indicated a negative impact to animal health (and even human health), and the upregulated *GLI1* expression positively correlated with diethanolamine may have reduced the meat tenderness of the bulls in GL16 because the *GLI1* gene can improve myogenesis.

The top 10 KEGG pathways with the largest number of genes and metabolites identified in this study were metabolic pathways, protein digestion and absorption, mineral absorption, bile secretion, the cAMP signaling pathway, linoleic acid metabolism, apoptosis, galactose metabolism, arachidonic acid metabolism, and the FoxO signaling pathway. Of these pathways, the metabolic pathways incorporated some metabolites and genes that were significantly correlated with each other. The leucine content showed a negative correlation with the *CYP3A4* and *FADS2* genes. Leucine, a type of branched-chain amino acid, is an essential amino acid that should be provided via the diet. Leucine modulates the signal pathways in muscle cells, which improves the protein synthesis in mammalian skeletal muscle [62]. Previous studies reported that a diet with high leucine supplementation decreased the feed intake and performance of pigs [63] and that adding an excessive content of isoleucine to the diet of fattening pigs could increase the IMF content without a change in other carcass traits [64]. *CYP3A4*, namely hepatic cytochrome P450 subfamily III A polypeptide 4, contributes to the metabolism of 45–60% of all drugs used in the clinical setting; its decreased expression and function were reported to be related to non−alcoholic fatty liver disease [65]. Other studies also reported decreased expression and activity of *CYP3A4* in both human liver tissue with steatosis or non−alcoholic fatty liver disease development [66,67], which indicated the significance of *CYP3A4* for liver health. In this study, leucine was upregulated in the livers of the GL16 group cattle compared to the YL16 group, which indicated that the *CYP3A4* gene was downregulated in the GL16 group. Thus, the decreased *CYP3A4* expression in the GL16 group might imply a negative impact on cattle health. The glycine content also showed a negative correlation with the *LDLRAD3* and *CLRN2* genes. Glycine, an abundant amino acid in cattle blood, was enhanced in GL16 compared with YL16. Glycine was reported to be associated with threonine and serine metabolism [68] and appears to be related to metabolic and chronic inflammatory conditions [69], the latter of which was reported to occur when cows were subjected to severe lipid mobilization [70]. The *LDLRAD3* gene was reported to be associated with de novo lipogenesis and protein kinase signaling by regulating SREBP, Sp1, and Ap4 [71]; this gene was also reported to play a critical role in cholesterol metabolism via cellular uptake of lipoproteins containing apolipoprotein E and B [72]. In this study, *LDLRAD3* and *CLRN2* were upregulated in YL16. Therefore, in total, the these upregulated genes and the decreased abundance of glycine indicated a favorable status for fat metabolism and the subsequent IMF deposition in the steers. For *CLRN2*, the exact function of this gene is unknown. In this study, even though we compared the transcriptomics and metabolomics data among three groups, we mainly focused on the comparison of the IMF deposition and its related metabolites and genes between the bulls and steers. In addition, for most of the DEGs such as *AKR1B10* and DFMs such as NG-dimethyl-L-arginine and pipecolinic acid, there is still little evidence to support their relationship with IMF deposition. More studies need to be conducted to reveal the biological connections between marbling and the relevant metabolites and genes.

## 5. Conclusions

This study demonstrated that the early castration of Holstein calves could be an applicable way to improve the beef marbling grade. In the untargeted metabolomics analysis, the pathways for protein digestion and absorption, biosynthesis of amino acids, alanine, aspartate and glutamate metabolism, and glycerophospholipid metabolism were enriched. The increased betaine, glycerol 3-phosphate, glutathione, acetylcarnitine, riboflavin, and alanine but decreased diethanolamine, glycine, and 2-hydroxyglutarate were the main newly found YL16-group-related metabolites; these probably contributed to the enhanced cattle health and subsequent beef quality. Based on a targeted metabolite analysis, early castration improved the fat content in the muscle. The mRNA expression levels of some genes such as *GLI1* and *NUF2* were decreased while that of *CYP3A4* was increased in the YL16 group, some of which were correlated with some beneficial bioactive metabolites in the liver. The early-castration treatment influenced the amino acids and related metabolic compounds by modifying the gene expression, which could be a beneficial strategy for the production of functional and flavorful beef products. Further studies need to be conducted to address the relationship of gene expressions and metabolites in and between various tissues such as the liver, muscle, and fat depots.

## Figures and Tables

**Figure 1 animals-12-03398-f001:**
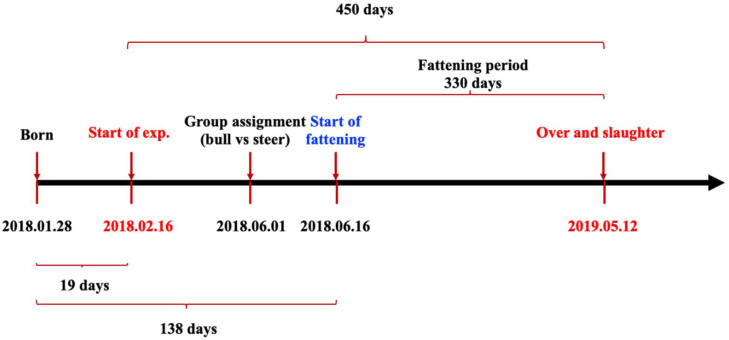
Timeline of the experimental design of this study.

**Figure 2 animals-12-03398-f002:**
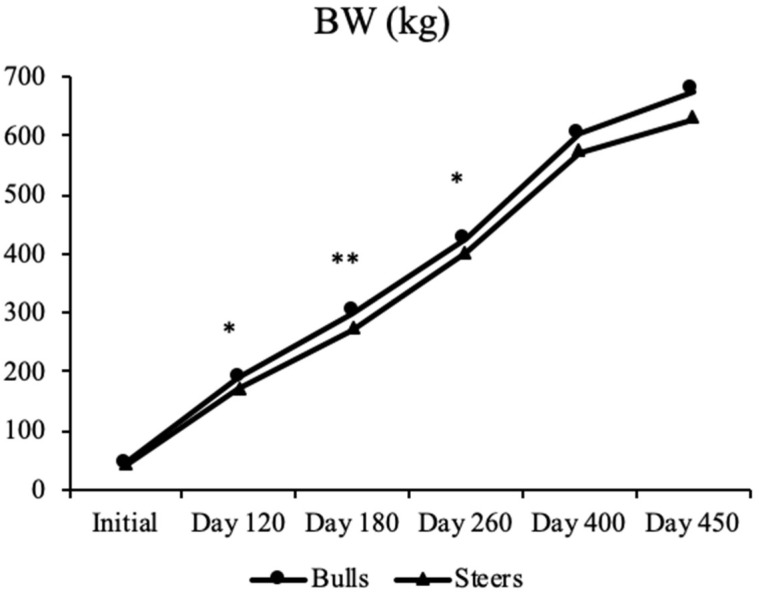
Comparison of BW between bulls and steers. * *p* < 0.05; ** *p* < 0.01.

**Figure 3 animals-12-03398-f003:**
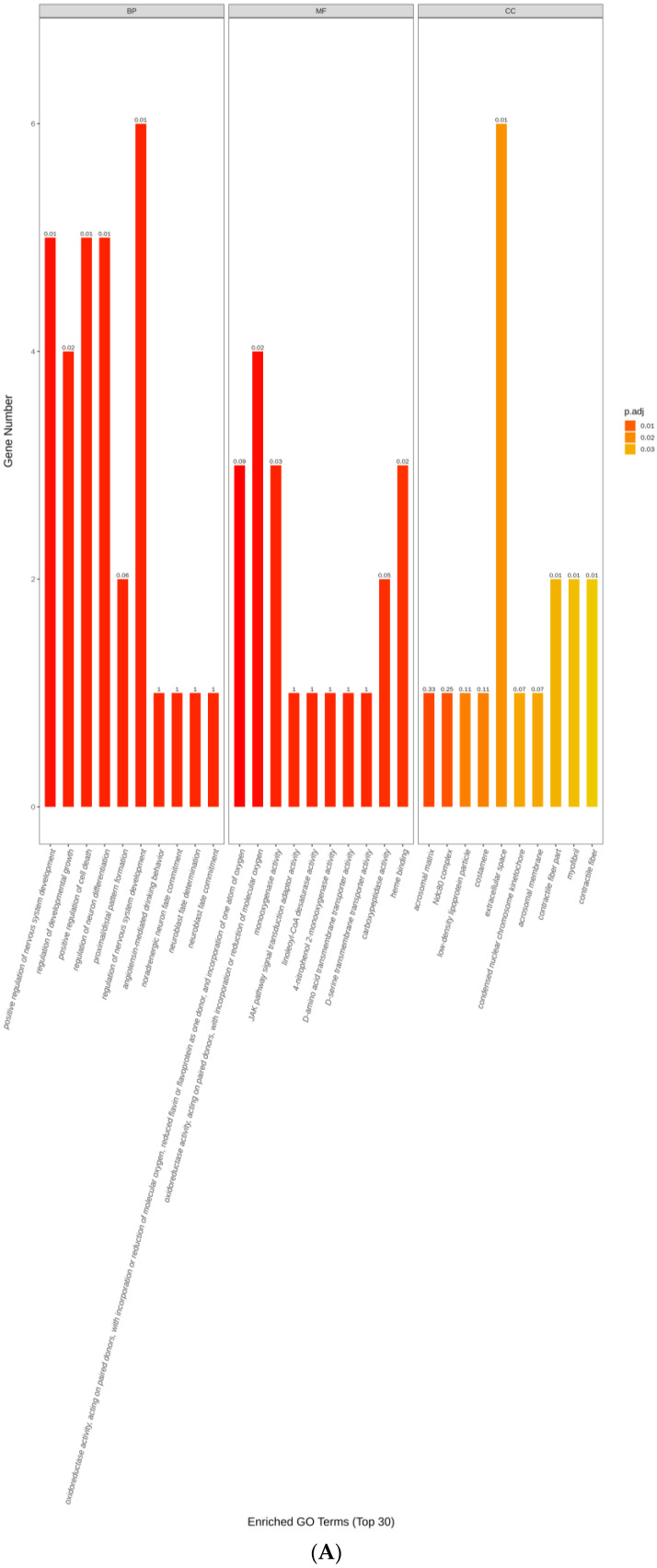
GO terms of different expressed genes for biological process, cellular components and molecular functions between GL16 and YL16 (**A**) or YL16 and YL26 (**B**). GL16 = non−castrated and slaughtered at 16 months of age; YL16 = castrated at birth and slaughtered at 16 months of age; YL26 = castrated at birth and slaughtered at 26 months of age.

**Figure 4 animals-12-03398-f004:**
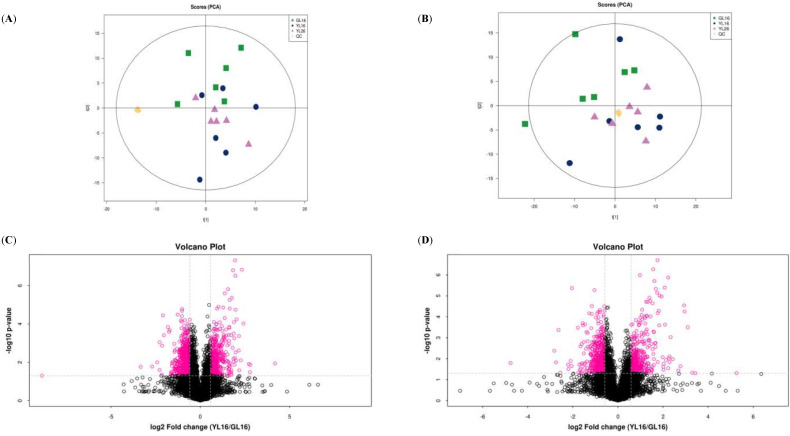
Metabolite profiling analysis of liver samples. Scatter plots of the PCA model based on all identified metabolite features of liver samples from the cattle in (**A**) positive mode and (**B**) negative mode, including the QC samples. Volcano plots of the comparison between GL16 and YL16: (**C**) positive mode, (**D**) negative mode; between YL16 and YL26: (**E**) positive mode, (**F**) negative mode; and between GL16 and YL6: (**G**) positive mode, (**H**) negative mode. GL16 = non−castrated and slaughtered at 16 months of age; YL16 = castrated at birth and slaughtered at 16 months of age; YL26 = castrated at birth and slaughtered at 26 months of age.

**Figure 5 animals-12-03398-f005:**
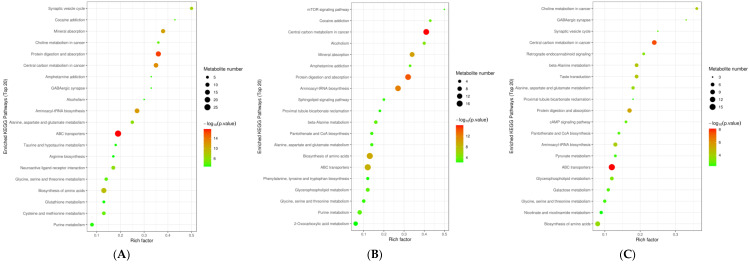
Top 20 enriched KEGG pathways of DFMs and possible pathways related to the fat metabolism in the liver: (**A**) GL16 vs. YL16; (**B**) YL16 vs. YL26; (**C**) GL16 vs. YL26. Rich factor = the ratio of the number of DFMs to total metabolites in each pathway; GL16 = non−castrated and slaughtered at 16 months of age; YL16 = castrated at birth and slaughtered at 16 months of age; YL26 = castrated at birth and slaughtered at 26 months of age.

**Figure 6 animals-12-03398-f006:**
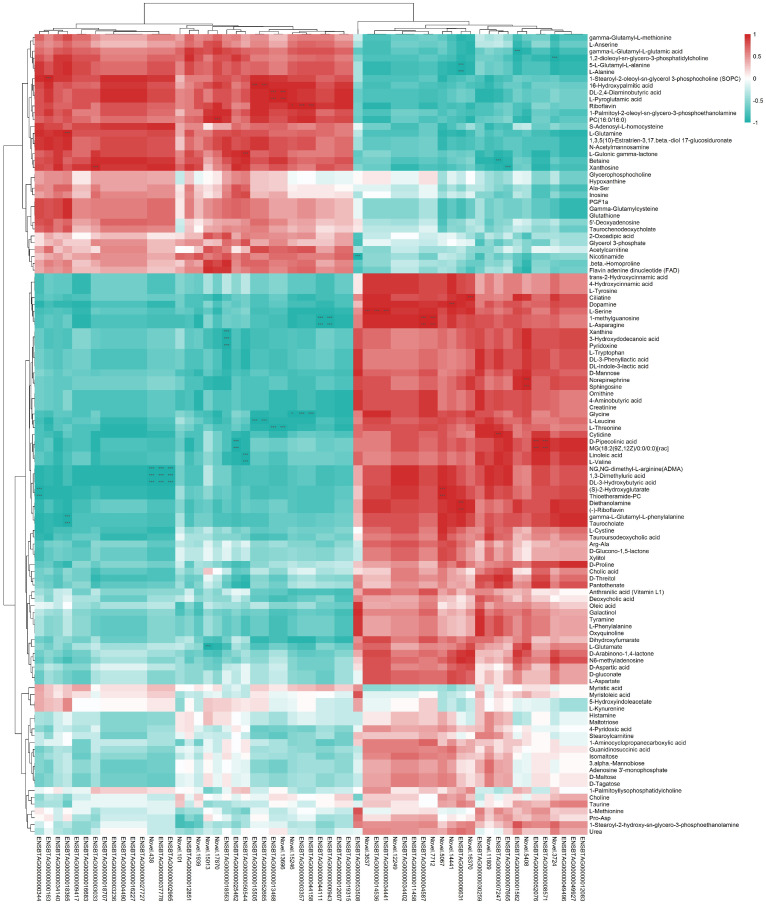
Pearson’s correlations between differentially expressed genes and differential metabolites in the comparison between GL16 and YL16. * *p* < 0.05; *** *p* < 0.001. GL16 = non−castrated and slaughtered at 16 months of age; YL16 = castrated at birth and slaughtered at 16 months of age.

**Table 1 animals-12-03398-t001:** Ingredient composition and nutrient levels of diets (DM basis).

Item	Suckling Period	Weaning Calf Period	Fattening Period		
	0–2.0 (m)	3.0–4.5 (m)	4.5–9.5 (m)	9.5–12.5 (m)	12.5–15.5 (m)
Ingredient composition (% of dry matter (DM))					
Steam-pressed corn	54.00	54.00	48.00	56.00	60.00
Puffed soybean flour	5.00	3.00	0.00	0.00	0.00
Soybean meal	25.00	22.00	20.00	10.00	8.00
Molasses	4.00	4.00	0.00	0.00	0.00
Cottonseed meal	0.00	0.00	0.00	2.00	2.00
DDGS	0.00	0.00	5.00	5.00	4.00
Rice bran	3.00	5.00	11.00	11.00	10.00
Whole corn silage	0.00	0.00	2.00	2.00	2.00 (0.00) ^3^
Corn stalk	0.00	0.00	10.00	10.00	10.00 (12.00) ^3^
Oat grass	2.50	5.00	0.00	0.00	0.00
Alfalfa hay	2.50	3.00	0.00	0.00	0.00
Premix ^1^	4.00	4.00	4.00	4.00	4.00
Total	100.00	100.00	100.00	100.00	100.00
Nutrient levels (% of DM) ^2^					
ME/(MJ/kg)	16.70	12.50	11.50	11.50	12.00
CP	18.50	17.50	16.50	14.00	12.50
EE	6.00	5.00	4.20	4.20	4.60
NDF	12.00	20.00	20.30	22.50	26.80
ADF	7.50	10.30	9.80	11.60	12.40
Ca	1.10	0.80	0.60	0.40	0.30
P	0.60	0.40	0.30	0.30	0.20

^1^ The premix provided the following per kg of diets: VA, 115 000 IU; VE, 750 mg; VB_5_, 450 mg; VB_12_, 0.8 mg; VK_3_, 130 mg; VB_9_, 10 mg; VD_3_, 3 000 IU; VE, 20 IU; VK_3_, 2 mg; Cu, 650 mg; Fe, 400 mg; Mn, 600 mg; Zn, 1000 mg; I, 45 mg; Co, 20 mg; Se, 35 mg. ^2^ ME was a calculated value, while the others were measured values. ^3^ Dietary formula for steers.

**Table 2 animals-12-03398-t002:** Effect of early castration on carcass characteristics of the Holstein cattle.

Item ^1^	Bulls	Steers	*p-*Value
Marbling score	0	2.5 ± 0.50	0.0001
Backfat thickness (mm)	4.7 ± 1.70	10.8 ± 2.36	0.001
LM area (cm^2^)	77.1 ± 6.49	68.6 ± 5.68	0.053

^1^ Marbling score: 1, devoid; 5, very abundant.

**Table 3 animals-12-03398-t003:** Effect of early castration on chemical and physio-chemical compositions of the Holstein cattle.

Item	Bulls	Steers	*p-*Value
Chemical composition (%)			
Moisture	64.1 ± 2.33	63.7 ± 1.59	0.75
Crude fat	6.8 ± 2.10	11.8 ± 3.24	0.008
Crude protein	23.2 ± 2.30	21.5 ± 0.82	0.17
Physio-chemical composition			
pH	5.5 ± 0.03	5.6 ± 0.03	0.131
CIE *L**	10.5 ± 2.30	12.0 ± 3.11	0.406
CIE *a**	14.7 ± 2.36	12.9 ± 0.53	0.157
Shear force (N)	63.2 ± 5.97	59.3 ± 2.83	0.544
Drip loss (%)	12.0 ± 1.40	10.6 ± 2.82	0.330
Cooking loss (%)	22.2 ± 3.29	19.5 ± 3.95	0.275

**Table 4 animals-12-03398-t004:** DFMs from liver samples in the key differential enriched KEGG pathways (false discovery rate < 0.05).

Metabolic Pathways	Metabolites
	GL16 vs. YL16
ABC transporters	Glutamate, alanine, glutathione, glutamine, glycerol 3-phosphate, leucine, riboflavin, betaine
Protein digestion and absorption	Glutamate, glycine, alanine, glutamine, tryptophan, tyrosine, valine
Biosynthesis of amino acids	S-adenosyl-L-homocysteine, glutamate, glycine, alanine, aspartate, glutamine
Glycerophospholipid metabolism	Glycerol 3-phosphate, choline, 1-stearoyl-2-oleoyl-sn-glycerol 3-phosphocholine (SOPC), glycerophosphocholine, giethanolamine
	YL16 vs. YL26
ABC transporters	Glutamate, alanine, arginine, glutamine, serine, glycerol 3-phosphate
Biosynthesis of amino acids	Glutamate, alanine, arginine, glutamine
Protein digestion and absorption	Glutamate, alanine, arginine, glutamine, serine, methionine
Purine metabolism	Adenosine 3’,5’-diphosphate (PAP), glutamine, ribose 5-phosphate
	GL16 vs. YL26
ABC transporters	Glutamate, aspartate, glutamine, phenylalanine, fructose, glycerol
Alanine, aspartate, and glutamate metabolism	Glutamate, aspartate, glutamine, 4-aminobutyric acid, aspartic acid
Protein digestion and absorption	Glutamate, aspartate, glutamine, phenylalanine, tyrosine, leucine
cAMP signaling pathway	Lactate, 4-aminobutyric acid, norepinephrine, 3-hydroxybutyric acid

GL16 = non−castrated and slaughtered at 16 months of age; YL16 = castrated at birth and slaughtered at 16 months of age; YL26 = castrated at birth and slaughtered at 26 months of age.

## Data Availability

The data that supported the findings of this study are available from the corresponding author upon reasonable request.

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
