# Peer review of "Multi-Omics Analysis of Transcriptomic and Metabolomics Profiles Reveal the Molecular Regulatory Network of Marbling in Early Castrated Holstein Steers"

_animals, 2022, doi:10.3390/ani12233398_

Round 1
Reviewer 1 Report
This manuscript provides an insight into the transcriptomic and metabolomics profiles of the liver tissue of Holstein bull and Holstein steer. The authors combined different approaches to find the candidate genes and metabolites. However, additional analysis needs to be performed to integrate metabolome and transcriptome data, the manuscript should be moderate revised, especially the Material and Methods section, the specific comments are list as below:
Line 35-36: The author could delete the description of used methods (such as: XXX was performed using Hiseq 2500).
Line 99: Why did author choose liver tissue instead of other tissue (such as muscle) in the present research, please clarify the reason in the introduction section or discussion section, besides, did the author compare the liver size of Holstein bull and Holstein steer?
Line 197:The identified metabolites should be annotated before the statistical analyses, please provide the detailed information of the used database for metabolites annotation.
Line 209: Provide the P value threshold for Fisher’s test.
Line 218: Please provide the detailed method for library preparation.
Line 220: Did the author used pair-ended strategy? PE150 or PE250?
Line 220: Please provide the detailed strategy for the quality control of raw data.
Line 222: Please provide the version of reference genome.
Line 229: Did the authors perform quality control before running DE analyses?
Line 229: Which method did the author choose for P value adjusting?
Line 232: Please provide the version of clusterProfiler, besides, clusterProfiler should be cited.
Line 285: It’s better put the GO enrichment figure in the manuscript instead of supplementary figure.
Line 336: From the PCA figure, none clear separate was found between the 3 groups, did the author try PCoA analysis?
Line 351: Please improve the quality of Figure 3, it’s hard to read the KEGG terms, the anthor could change the font size and reduce the numbers of terms.
Line 389: It can be inferred from the PCC figure that strong correlation exists between the DE metabolites and DE genes, it will be better if the author could draw a correlation network based on the PCCs.
Line 514: The title showed that the manuscript is focused on the multi-omics analysis, however, only minority of the research work was performed on the integrated analysis, more analysis work should be done to fulfill the present research, for example, identifying the metabolites/genes with the highest correlation number with other DE candidates.
Author Response
This manuscript provides an insight into the transcriptomic and metabolomics profiles of the liver tissue of Holstein bull and Holstein steer. The authors combined different approaches to find the candidate genes and metabolites. However, additional analysis needs to be performed to integrate metabolome and transcriptome data, the manuscript should be moderate revised, especially the Material and Methods section, the specific comments are list as below:
Response: Thank you for your all valuable comments.
Line 35-36: The author could delete the description of used methods (such as: XXX was performed using Hiseq 2500).
Response: As you suggested, we revised the description appropriately. (Line 35-36)
Line 99: Why did author choose liver tissue instead of other tissue (such as muscle) in the present research, please clarify the reason in the introduction section or discussion section, besides, did the author compare the liver size of Holstein bull and Holstein steer?
Response: As you suggested, we already added the relevant reason about selecting the liver tissue as target sample in Introduction Section. (Line 112-117)
Besides, unfortunately, we did not compare the liver size of two groups. But, according to Wang et al., 2017, the difference in liver energy metabolism, fatty acid synthesis and regulatory gene expression among cattle breeds may lead to different IMF contents. Therefore, we supposed that the difference of both energy and fat metabolism in the liver of cattle from two groups would cause different IMF contents in this study.
[Reference] Wang B., Fu X., Liang X., Wang Z., Yang Q., Zou T., Nie W., Zhao J., Gao P., Zhu M., de Avila J., Maricelli J., Rodgers B., and Du M. Maternal Retinoids Increase PDGFRα+ Progenitor Population and Beige Adipogenesis in Progeny by Stimulating Vascular Development. EBioMedicine, 2017, 18:288-299.
Line 197:The identified metabolites should be annotated before the statistical analyses, please provide the detailed information of the used database for metabolites annotation.
Response: we already added the relevant information in Materials and methods section. (Line 230-233)
The quantity was 30000+, and 6000+ metabolites were self-built with standard products including 4000+ of human and animal and 2000+ of plant. The others were from four public databases: Mass Bank, Metlin, HMDB and MoNA corresponding secondary standard spectrum library.
Line 209: Provide the P value threshold for Fisher’s test.
Response: As you suggested, we added the P value threshold in Materials and Methods section. (Line 251)
Line 218: Please provide the detailed method for library preparation.
Response: As you suggested, we already added the relevant information in Materials and Methods section. (Line 261-272)
Line 220: Did the author used pair-ended strategy? PE150 or PE250?
Response: Yes, we used PE150.
Line 220: Please provide the detailed strategy for the quality control of raw data.
Response: We added the relevant contents in Materials and Methods section. (Line 276-282)
Raw data (or Raw reads) of fastq format were firstly processed through in-house perl scripts. In this step, remove the adapter sequence and filter out low quality (low quality, the number of lines with a string quality value less than or equal to 25 accounts for more than 60% of the entire reading) and N (N means that the base information cannot be determined) ratio is greater than 5% reads to obtain clean reads that can be used for subsequent analysis.
Line 222: Please provide the version of reference genome.
Response: We already added the relevant contents in Materials and Methods section. (Line 282-284)
Then clean reads were separately aligned to reference genome with orientation mode using HISAT2 software (http://daehwankimlab.github.io/hisat2/) to obtain mapped reads.
Line 229: Did the authors perform quality control before running DE analyses?
Response: For the quality control of raw data, we performed quality control. But, for count of DE analyses, we did not perform QC.
Line 229: Which method did the author choose for P value adjusting?
Response: We revised the original sentence.
Line 232: Please provide the version of clusterProfiler, besides, clusterProfiler should be cited.
Response: We made a mistake for this package. We used topGO package, not clusterProfiler, and we added the relevant information in Materials and Methods section. (Line 296)
Line 285: It’s better put the GO enrichment figure in the manuscript instead of supplementary figure.
Response: As you suggested, we already moved the GO enrichment figure (original Figure S3) from supplementary data to manuscript, and marked it in Figure 2.
Line 336: From the PCA figure, none clear separate was found between the 3 groups, did the author try PCoA analysis?
Response: We did not try the PCoA analysis.
Line 351: Please improve the quality of Figure 3, it’s hard to read the KEGG terms, the anthor could change the font size and reduce the numbers of terms.
Response: Thank you for your suggestion. We already put the much more clear figure at the bottom of the article, and now marked as Figure 5.
Line 389: It can be inferred from the PCC figure that strong correlation exists between the DE metabolites and DE genes, it will be better if the author could draw a correlation network based on the PCCs.
Response: Unfortunately, we can not do this work. But, we will try this technique next time.
Line 514: The title showed that the manuscript is focused on the multi-omics analysis, however, only minority of the research work was performed on the integrated analysis, more analysis work should be done to fulfill the present research, for example, identifying the metabolites/genes with the highest correlation number with other DE candidates.
Response: Thank you for your valuable comment. In this study, even though we analyzed and compared the transcriptomics and metabolomics data among three groups, it is mainly focused on the comparison in IMF deposition and its related metabolites and genes between bulls and steers. Therefore, for experiment results, especially correlation analysis data, it was mainly discussed between GL16 and YL16. We already added more contents and reflected them in Discussion section. (Line 791-794)

Reviewer 2 Report
The authors have addressed an important area of study on how the transcriptome and metabolome of castrated and uncastrated steers/bulls differ, and how this might affect their ability to produce beef with IMF (marbling). However, the authors need to work more on emphasizing how and to which extent they have achieved their main objective(s). Also, the analysis of transciptomic and metabolomic data is done in a descriptive manner, whereas a more focused approach on trying to compare the same using animals with high marbling and low marbling score would have revealed more relevant and arguably more interesting results.
General writing needs to be improved (details provided as comments in the attached pdf. Each highlighted sentence/section carries a comment with regard to what the issue is/how it can be fixed).
Bioinformatic analysis and figure legends need to be improved (details provided as comments in the attached pdf)
Discussion and the conclusion should be more focused. Try to emphasize on how transcriptome and metabolome data might actually directly correlate to the level of marblling observed in the animals (details provided as comments in the attached pdf). Furthermore, a discussion on what the findings might imply interms of animal disease pathology would be far more relevant than trying to make vague implications on its effect on human health (as evident in the text).

Author Response
The authors have addressed an important area of study on how the transcriptome and metabolome of castrated and uncastrated steers/bulls differ, and how this might affect their ability to produce beef with IMF (marbling). However, the authors need to work more on emphasizing how and to which extent they have achieved their main objective(s). Also, the analysis of transcriptomic and metabolomic data is done in a descriptive manner, whereas a more focused approach on trying to compare the same using animals with high marbling and low marbling score would have revealed more relevant and arguably more interesting results.
Response: Thank you very much for sparing your valuable time to review this manuscript, and providing your valuable comment. In this study, even though we analyzed and compared the transcriptomics and metabolomics data among three groups, it is mainly focused on the comparison in IMF deposition and its related metabolites and genes between bulls and steers. Therefore, for experiment results, such as correlation analysis data, it was mainly discussed between GL16 and YL16 in order to avoid redundant contents.
Comment : General writing needs to be improved (details provided as comments in the attached pdf. Each highlighted sentence/section carries a comment with regard to what the issue is/how it can be fixed).
Response: Thank you very much for all your valuable comments. We already revised all the parts that you pointed out, and highlighted as follows.
- “Increased betaine, alanine and glycerol 3-phosphate were highly found in YL16 group” was changed to “Asignificant increase in the presence of betaine, alanine and glycerol 3-phosphate was observed in YL 16 group”. (Line 38-40)
- “Compared to the GL16 and YL26 groups, enhanced glutathione, acetylcarnitine, and riboflavin, but decreased diethanolamine and 2-hydroxyglutarate, were the main newly identified YL16 group-related metabolites” was changed to “Compared to the GL16 and YL26 groups, a significant increase in the presence of glutathione, acetylcarnitine, and riboflavin, but decrease in diethanolamine and 2-hydroxyglutarate, were identified in YL16 group”. (Line 41-43)
- “have been becoming” was changed to “have become increasingly”. (Line 80)
- “RNA-Seq technique” was changed to “bulk RNA-Seq”. (Line 92)
- “as the cost of extensive analysis of the linkage between metabolite biosynthesis and gene expression is acceptable” was changed to “cost-effective methods of analyzing the linkage between metabolites and gene expression emerge”. (Line 104-105)
- “combination of metabolomics and transcriptomics” was changed to “combination of metabolomics and transcriptomics data”. (Line 105-106)
- Thankyou for your comment. We already deleted the original sentence “For example, insulin signaling and nuclear FoxO1 signaling pathways were found to be differentially enriched in castrated- and uncastrated-cattle following castration ”, and revised this part as follows:
Transcriptomics provides an effective tool to explore genes that might contribute to IMF deposition. Marbling has diverse highly regulated metabolic pathways that can play significant roles in their production under castration. Metabolomics has also been used in IMF metabolism and biomarker identification [Chen et al., 2022]. Metabolites that are the final products of diverse biological metabolism serve as precise biomarkers that reflect upstream biological processes including environmental changes and genetic mutations [Nicholson and Lindon, 2008]. However, metabolomic biomarkers are known to might have low reproducibility due to sample sources, population heterogeneity, parameters setting in the metabolomics data, and different experiment protocols, etc [Huang et al., 2016]. Currently, as system biology and bioinformatics tools develop, and plus, cost-effective methods of analyzing the linkage between metabolites and gene expression emerge, the combination of metabolomics and transcriptomics data provides a powerful approach for gaining a comprehensive insight into the mechanisms of IMF deposition responses to castration at the cellular and molecular levels than either approach alone. This new approach may observe fat deposition from an aspect of system biology and enhance the credibility of biomarkers [Yang et al., 2017]. Therefore, comprehensive analysis of the metabolomics and transcriptomics can provide a better understanding of marbling responses to castration. (Line 92-111)
- “integrated transcriptome and metabolome in liver tissues were analyzed to evaluate” was changed to “integrated analysis of transcriptomic and metabolomic data was used to evaluate”. (Line 120)
- “were analyzed in order to provide a better understanding” was changed to “was used to gain a better understanding”. (Line 122)
- “provided” was changed to “provides”. (Line 125)
- We already revised the original sentence, thus, the “adaptation” was deleted.
- As you suggested, “Table 2. Effect of early castration on growth performance of Holstein cattle” was converted to figure, and marked as Figure 2. We put the Figure 2 at the bottom of this manuscript temporarily.
- Considering the inappropriate content, we decided to delete the original “The correlation of transcript expression between samples is the most important indicator to test the reliability of experiment results and the rationality of sampling. ”.
- “different expressed genes” was changed to “differentially expressed genes”. (Line 448)
- As you suggested, we already labelled only 10 Up/Down regulated genes, and we made the background in white color inFigure S2. Also, we revised the figure legend appropriately.
- Wealready revised the panel A in Figure 3, and put a new one. In this study, we used topGO package (Version topGO: 2.46.0; GO.db: 3.14.0) for GO function enrichment and KEGG pathway enrichment analysis, not clusterProfiler package. We already revised some figure legend, e.g. the legend of Figure 5. If there are still some inappropriate legends of table or figure, please point out them. Thank you for your valuable suggestion.
- “differential genes” was changed to “differentially expressed genes”. (Line 477)
- “PCA score plots” was changed to “PCA plots”. (Line 488)
- “alleviate” was changed to “alleviating”. (Line 574)
- “growing period afterward” was changed to “subsequent growth period”. (Line 576)
- “scientific experimental basis” was changed to “scientific basis”. (Line 578)
- “whole growth periods” was changed to “entire growth period”. (Line 580)
- “of animal” was changed to “of the animal”. (Line 582)
- “present results” was changed to “the results of the current study”. (Line 607)
- “The finding in this study is agreement” was changed to “This finding is in agreement”. (Line 608)
- “the number of multiply mapped reads” was changed to “the number of reads with multiple matches”. (Line 619)
- “0.96” was changed to “a correlation coefficient of 0.96”. (Line 621)
- “an” was changed to “a”. (Line 623)
- “nutrients” was changed to “nutrient”. (Line 623)
- “under early castration condition” was changed to “in early castrated animals”. (Line 628)
- “trial” was changed to “study”. (Line 630)
- We already re-arranged the sentence as follows: GO enrichment analysis between GL16 and YL16 showed that DEGs involved in biological processes were mainly implicated in positive regulation of nervous system development and regulation of developmental growth, and the related genes (GLI1, IRX3, CUX2, DUSP6 et al.) were down-regulated in early castrated animals, indicating castration imparts negative impact to animal growth performance. This finding is similar to previous study that the castration restricted muscle growth and further body weight in mouse [29]. (Line 625-631)
- “Previous study” was changed to “A previous study”. (Line 638)
- “other” was changed to “another”. (Line 640)
- “KEGG analysis of this study” was changed to “Results of the KEGG analysis from the current study”. (Line 643-644)
- We re-arranged the sentence as follows: In these two lipid metabolism related pathways, CYP2E1 and CYP3A4 were detected, and the expression of two genes were also up-regulated in YL16 group. (Line 655-657)
- “identified the enriched pathway of” was changed to “uncovered enriched pathways for”. (Line 658)
- We deleted the “the”, and now that is “glycerol 3-phosphate”. (Line 657)
- “glycerol 3-phosphate and alanine” was changed to “biosynthesis of glycerol 3-phosphate and alanine”. (Line 661)
- We revised the original sentence to “Furthermore, using the LC-MS/MS untargeted metabolomics, other metabolites associated with the meat function or sensory traits were detected.” (Line 664-665)
- As you suggested, we already deleted the repeated sentence “which were GL16 group-related increased metabolites.”, and the arranged sentence is “For the GL16 group-related DFMs, diethanolamine and 2-hydroxyglutarate are potentially toxic and deleterious compounds.” (Line 691-692)
- “liver and kidney” was changed to “liver/kidney”. (Line 695)
- “improvement” was changed to “upregulation”.(Line 726)
- “of” was changed to “in”. (Line 728)
- We changed the “the enhanced content of betaine in YL16 may promote fat deposition in the muscle” to “the enhanced content of betaine in YL16 may activate fat metabolism in the body”. (Line 739-740)
- We changed “the This study demonstrated that implement of early castration to calves could be a practicable way to improve beef marbling grade” to “This study demonstrated that treatment of early castration to Holstein calves could be an applicable way to improve the beef marbling grade”. (Line 799-800)
- We revised the original sentence to “In the untargeted metabolomics analysis, the pathway of protein digestion and absorption, biosynthesis of amino acids, alanine, aspartate and glutamate metabolism, and glycerophospholipid metabolism were enriched,”. (Line 800-803)
- We revised the original sentence to “and the increased betaine, glycerol 3-phosphate, glutathione, acetylcarnitine, riboflavin, and alanine, but decreased diethanolamine, glycine and 2-hydroxyglutarate, were the main newly found YL16 group-related metabolites, which probably contribute to the enhanced cattle health and subsequent beef quality.”. (Line 803-806)
Comment: Bioinformatic analysis and figure legends need to be improved (details provided as comments in the attached pdf)
Response: We already revised some figure legend, e.g. the legend of Figure 5.
Comment: Discussion and the conclusion should be more focused. Try to emphasize on how transcriptome and metabolome data might actually directly correlate to the level of marblling observed in the animals (details provided as comments in the attached pdf). Furthermore, a discussion on what the findings might imply interms of animal disease pathology would be far more relevant than trying to make vague implications on its effect on human health (as evident in the text).
Response: Thank you for your valuable comments. As you suggested, we tried to connect the transcriptome and metabolome data with IMF deposition, namely marbling of animal, according to the comments in the attached PDF. Also, we tried to discuss and connect the findings with animal health throughout the Discussion section, not with human health as far as possible.
